# MultiDefectNet: Multi-Class Defect Detection of Building Façade Based on Deep Convolutional Neural Network

**Kisu Lee [1], Goopyo Hong [2], Lee Sael [3] 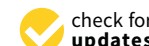, Sanghyo Lee [4],* and Ha Young Kim [1],***

[1] Graduate School of Information, Yonsei University, 50 Yonsei-ro, Seodaemun-gu, Seoul 03722, Korea; kisu0928@naver.com

[2] Division of Architecture and Civil Engineering, Kangwon National University, 346 Jungang-ro, Samcheok-si, Gangwon-do 25913, Korea; goopyoh@kangwon.ac.kr

[3] Department of Data Science, Ajou University, 206 World Cup-Ro, Yeongtong-gu, Suwon-si, Gyeonggi-do 16499, Korea; sael@ajou.ac.kr

[4] Division of Smart Convergence Engineering, Hanyang University ERICA, 55 Hanyangdaehak-ro, Sangnok-gu, Ansan-si, Gyeonggi-do 15588, Korea

* Correspondence: mir0903@hanyang.ac.kr (S.L.); hayoung.kim@yonsei.ac.kr (H.Y.K.); Tel.: +82-31-400-5965 (S.L.); +82-2-2123-4194 (H.Y.K.)

**Abstract:** Defects in residential building façades affect the structural integrity of buildings and degrade external appearances. Defects in a building façade are typically managed using manpower during maintenance. This approach is time-consuming, yields subjective results, and can lead to accidents or casualties. To address this, we propose a building façade monitoring system that utilizes an object detection method based on deep learning to efficiently manage defects by minimizing the involvement of manpower. The dataset used for training a deep-learning-based network contains actual residential building façade images. Various building designs in these raw images make it difficult to detect defects because of their various types and complex backgrounds. We employed the faster regions with convolutional neural network (Faster R-CNN) structure for more accurate defect detection in such environments, achieving an average precision (intersection over union (IoU) = 0.5) of 62.7% for all types of trained defects. As it is difficult to detect defects in a training environment, it is necessary to improve the performance of the network. However, the object detection network employed in this study yields an excellent performance in complex real-world images, indicating the possibility of developing a system that would detect defects in more types of building façades.

**Keywords:** multi-class defect detection; building façade defect; deep learning; Faster R-CNN

## 1. Introduction

A building should exhibit good performance in supporting the activities of its occupants. Among the various types of buildings, residential buildings should perform particularly well because occupants spend most of their time in them [1]. Hence, it is crucial to minimize defects to maintain the performance of residential buildings [2,3]. In particular, building façades are considered essential elements of buildings because they influence the appearance, structural safety, and insulation of the buildings, but also play the role of an exterior shield against weather and pollution [4]. However, continuous exposure to poor environmental conditions during a long service life accelerates aging relatively faster compared with other building components [5]. This phenomenon is eventually manifested in various types of defects on the building façade [6]. If various defects in the building façade are ignored, they may result in shortening the service life, damage to appearance, and increased

maintenance costs [7]. Ultimately, it is ideal to prevent all types of defects in the design or construction stages, but this is a very difficult goal to achieve. Thus, there is a need for a method to effectively monitor defects in the maintenance phase and actively respond to the occurrence of the defects [8,9].

However, traditional defect management is associated with various issues, such as the subjectivity of the results arising from human-centered inspection, time consumption, occurrence of human causalities, and an increase in labor costs [10]. Therefore, it is necessary to develop a technology that can continually and automatically monitor defects in residential buildings that minimize the dependence on manpower [11–13]. Furthermore, there are various types of defects in residential buildings [14,15], and each defect type in the real world appears in an irregular pattern [10,16]. To consider the characteristics of these defects, automated defect monitoring technology should be able to simultaneously detect and effectively classify various types of defects in image data.

Deep learning techniques are data-driven methods that do not require rules. The process of building a model only needs to select a suitable network structure, a function to evaluate the model output, and a reasonable optimization algorithm [10]. Deep learning techniques are driving advances in computer vision to tackle the drawbacks of classical defect detection models that allow the automatic capture of intricate structures of large-scale data with models comprising multiple processing layers [17]. Several previous studies have attempted to apply deep-learning methods to detect cracks in various structures and defects in sewer pipes [18–21]. However, as the residential building façade is designed in various ways, we should be able to simultaneously classify various types of defects that appear in complex backgrounds.

From this perspective, in this study, we propose a multiclass object detection model capable of distinguishing various types of defects occurring in residential building façades by employing a faster region proposal convolutional neural network (Faster R-CNN).

## 2. Literature Review

As the building ages owing to various factors, it is crucial to continually inspect its various defects. From this perspective, several studies have been conducted to identify an efficient building inspection plan [8,22–24]. Kim et al. [8] presented a probabilistic approach to establish an optimum inspection/repair strategy for Reinforced Concrete structures subjected to pitting corrosion. Liao [22] proposed a method for the development of inspection strategies for the construction industry. Pires [23] presented a method to facilitate performance assessment and analyzed the degradation of building envelopes with a focus on painted finishes. Bortolini et al. [24] presented a building inspection system to evaluate the technical performance of existing buildings based on the consideration of the entire buildings and the interdependence of their parts. Based on a review of literature on these topics, it has been shown that the majority of the publications focused on inspection strategies or maintenance plans. However, as described above, the traditional human resource-oriented inspection has its own limitations. Thus, it is necessary to develop a technology that can continually and automatically monitor the building defects.

To this end, several studies proposed structural health monitoring (SHM) techniques. In general, the SHM system employs a vibration-based structural system identification technique using a numerical method [25–27]. Rabinovich et al. [25] developed a robust computational tool based on a combination of the extended finite element method (XFEM) and genetic algorithm (GA) to accurately detect and identify cracks in two-dimensional structures. Chatzi et al. [26] improved the XFEM-GA detection model by adding a novel genetic algorithm that accelerates the convergence of the scheme, and a generic XFEM formulation of an elliptical hole. Cha et al. [27] proposed a hybrid multi-objective genetic algorithm as a damage detection method to solve inverse problems to minimize the difference in the modal strain energy (MSE) in each structural element. However, fundamentally, the SHM system has various limitations, such as cost issues, compensation for environmental impacts, and noise signals, because of the installation of multiple sensors. Further, as the SHM system monitors only the

structural damage, it has a drawback in that it cannot detect various types of defects, such as cracks, water leakage, detachment, corrosion, and efflorescence [10,27].

Therefore, several studies have been conducted on defect detection methods based on image processing techniques (IPT) [28–30]. Laofor et al. [28] presented a defect detection and quantification system to augment subjective visual quality inspections in architectural work based on the specification of defect positions and the quantification of defect values. This method is able to use defect feature analysis to quantify the defect value from digital images using a digital image processing technique. Liao et al. [29] proposed a digital image recognition algorithm that consisted of three different detection techniques: K-means in H, DCDR in RGB, and DCDR in H, to improve the detection accuracy of the rusted areas on steel bridges. Shen et al. [30] developed a Fourier-transform-based steel bridge coating defect-detection approach (FT-DEDA) that makes use of the fact that the differences between background pixels are not as large as the differences between defect pixels to detect their existence. However, as image data obtained in the real world are quite diversified, IPT using prior knowledge are limited in recognizing defects in image data [27].

The deep learning technique been has intensively researched in the field of image recognition and can address these issues of IPT. In the construction field, there are several studies that have used deep learning to monitor defects in civil structures [18–21]. Li et al. [18] proposed an automated defect detection and classification method from closed-circuit television (CCTV) inspections based on a deep convolutional neural network (DCNN) that takes advantage of the large volume of inspection data. Dung et al. [19] proposed a vision-based method for concrete crack detection and density evaluation using a deep fully convolutional network (FCN). Yang et al. [20] developed a transfer learning method based on multiple DCNN knowledge for crack detection. Yin et al. [21] proposed an automated defect detection system with an object detector based on a convolutional neural network (CNN), commonly known as the YOLOv3 network. There are differences among civil structures and buildings as documented on previous studies depending on the characteristics of the facilities. That is, as the building façade is made up of various shapes, it is critical to recognize specific defects within image data. Furthermore, as the types of defects are diverse and their shapes are irregular, there is a need for a model that can simultaneously classify various types of defects.

From this point-of-view, in this study, we propose a multiclass object detection model that can simultaneously recognize various defects in image data obtained from various building façades.

## 3. Research Methodology

For the multiclass object detection model based on deep learning, several models have been proposed, starting with regions with CNN features (R-CNN) [31]. These models are mainly classified into R-CNN two-stage and YOLO one-stage detectors. In this study, by employing the Faster R-CNN model (two-stage detectors) proposed by Ren et al. [32] as a base model, we simultaneously detected multiple defects in single image data. Although the inference time of Faster R-CNN was slower than that of one-stage detectors, Faster R-CNN yielded a better performance in terms of accuracy. Thus, in this study, we selected the Faster R-CNN as the base model.

### 3.1. Overall Architecture

In this study, we developed a model that is capable of multiclass defect detection from real-world image data obtained from various building façades by employing the Faster R-CNN structure as the base model, as described above. As shown in Figure 1, Faster R-CNN is largely composed of three modules. Feature maps are extracted from the raw image data through the first module, the shared CNN. The feature map that is the output of the shared CNN is used as the input of the second module, the region proposal network (RPN). The RPN proposes regions where an object is likely to exist in an image. The third module is a fast R-CNN detector that takes the outputs of the previous two modules as inputs and predicts the location of the object and the class of the object. As observed in Figure 1, all modules are trained end-to-end. The important point is that the overall framework takes on a type

of attention mechanism because RPN is responsible for the suggestion of the location that the Fast R-CNN detector mainly focuses on.

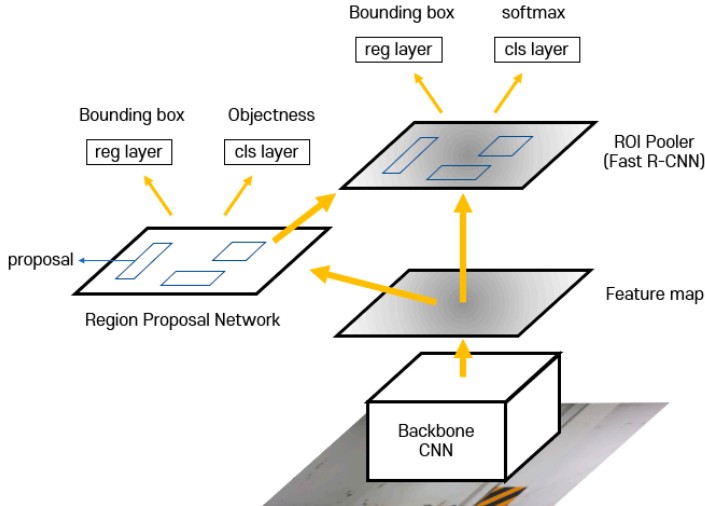

**Figure 1.** Overall structure of the faster regions with convolutional neural network (Faster R-CNN).

### 3.2. Shared Convolutional Neural Network

The shared convolutional neural network (CNN), the first module of Faster R-CNN, is the backbone of the entire structure, and is responsible for extracting features from raw image data. The feature map is the output extracted from the backbone and is used as the input of the next module.

The shared CNN can use multiple CNN models such as VGG16 [33], ResNet [34], and ResNext [35]. Depending on the structure of the backbone, the number of layers and computational costs may vary. In deep learning, it is well-known that with an increase in the number of layers, better features are extracted from raw data. However, an increase in the number of layers translates into high-computational costs and additional parameters. As the layer becomes deeper, vanishing gradient problem is highly likely to occur when performing backpropagation calculations. Additionally, overfitting is also highly likely to occur when training data are insufficient with an increase in the number of parameters. Accordingly, it is crucial to set up an appropriate backbone depending on various situations, such as the number of datasets.

According to a previous study, the COCO's standard metric mAP@[0.5, 0.95] for the MS COCO dataset improved by 6% compared with VGG16 when ResNet-101 was used as the backbone of Faster R-CNN. Therefore, in this study, ResNet was selected as the backbone model. ResNet-50, a lighter model than ResNet-101, was used to prevent overfitting depending on the number of the datasets.

As shown in Figure 2, ResNet is a deep convolutional network (DCNN) that employs a residual learning method with a shortcut connection (skip connection) to alleviate the vanishing gradient problem that occurs when the number of the layers of the neural network (NN) increases.

### 3.3. Feature Pyramidal Network

Feature pyramids can be viewed as a tool for scale-invariant properties in object detection. That is, it is a method used to detect objects of various sizes in one image. In previous studies [36–38], before the feature pyramidal network (FPN) was proposed, feature pyramids were generated while the raw image was resized to detect objects of various sizes. However, these were very expensive processes in terms of computational costs and memory usage. Therefore, there was a tendency not to use them in deep learning object detectors, and FPN complements this previous method [39].

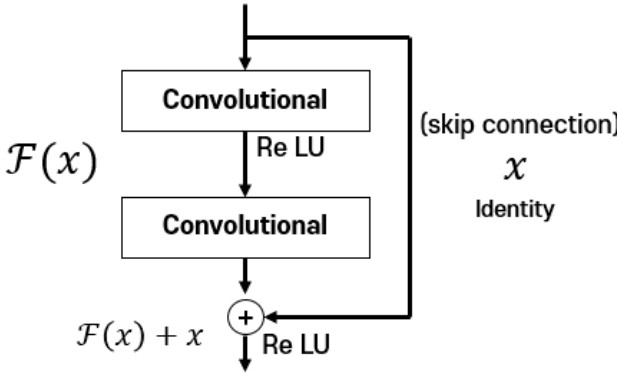

**Figure 2.** Structure of a building block in residual learning.

As shown in Figure 3, FPN consists of the bottom-up pathway, top-down pathway, and lateral connections. The bottom-up pathway is the feed-forward computation process of the backbone CNN. These outputs feature maps of multiple scales and assign layers with the same output feature map size at the same network stage. A single pyramidal level is defined for each stage as the output of the last layer in each stage is a high-level representation in each stage. This is selected as the reference set of the feature maps. For ResNet, a layer block that outputs a feature map of the same size by adjusting the stride is composed, and the output of the last layer of each block is used as a pyramid. However, the 1st block is excluded because the pyramidal memory is large.

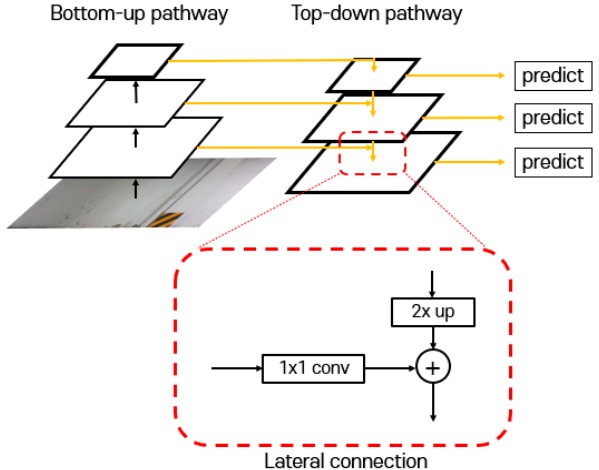

**Figure 3.** Structure of feature pyramidal network (FPN).

The top-down pathway makes a high-level representative feature map into a higher-resolution image using upsampling. The loss of location information arising from the upsampling process is compensated by using lateral connections with the feature map in the bottom-up layer. The output of the FPN is used as the input of the RPN and region-of-interest (RoI) pooler, and facilitates the prediction of the bounding box of various sizes.

*3.4. Region Proposal Network*

As seen in Figure 4, the region proposal network (RPN) refers to a network that generates region proposals by taking the feature map, which is the output of the backbone CNN, as an input. Region proposals are a set of rectangular object proposals, and objectness scores are calculated for each proposal. Region proposals are generated by sliding a small network (sliding window) over the feature map (shared CNN output). This small network takes the n × n spatial window of the feature map as an input and maps it onto the lower-dimensional feature map.

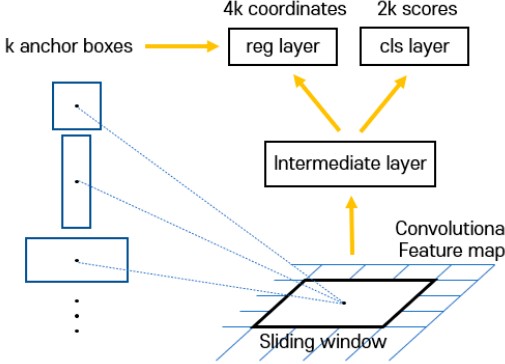

**Figure 4.** Structure of region proposal network (RPN).

The output features through the small network (sliding window) are fed into two fully connected sibling layers, namely the box-regression layer (reg) and box-classification layer (cls), respectively. Herein, n, the size of the sliding window, is a hyperparameter, and is set to three (i.e., n = 3), as in the previous study.

Meanwhile, an anchor box is set at the center of each sliding window position. This is to simultaneously predict multiple region proposals at one position, and the number of anchor boxes is a hyperparameter. Once the scale and aspect ratio of the anchor box are determined, the anchor boxes are generated. These are equal to (the number of scales × the number of ratios). If the number of anchor boxes is k and the feature map size is W × H, the total number of anchors is WHk. That is, the entire pixel of the feature map becomes the center of the sliding window, and an anchor box is generated at each center. In this study, three scales and aspect ratios were used, respectively, that is, k = 9. The output from each anchor box was used as the input of two sibling layers through the intermediate layer. The box regression layer derives 4 k outputs by encoding the coordinates of the box for each anchor box, while the box-classification layer has 2 k outputs (object or not) because it only determines the objectness for each anchor box.

The loss of the RPN is a value obtained by normalizing the sum of the loss calculated for each anchor. A binary class label (object or not) is assigned to each anchor, and the assignment criterion is the intersection over union (IoU) with the ground-truth box. IoU is defined in Figure 5.

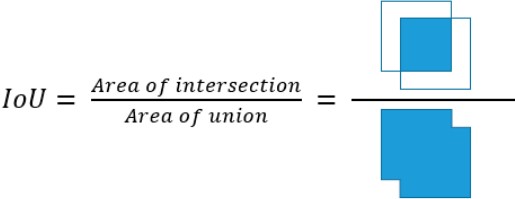

$$IoU = \frac{Area\ of\ intersection}{Area\ of\ union} = $$

**Figure 5.** Definition of the Intersection over Union(IoU).

The anchor having the highest IoU with one ground-truth box is defined as positive label 1. Of the nonpositive label anchors, if the IoU with all ground-truth boxes is less than 0.3, it is defined as a negative label 0. Herein, anchors that do not correspond to both label criteria are excluded from training. The loss function built using this definition is shown in Equation (1).

$$\mathcal{L}(\{p_i\},\ \{t_i\}) = \frac{1}{N_{cls}} \sum_i \mathcal{L}_{cls}\left(p_i,\ p_i^*\right) + \lambda \frac{1}{N_{reg}} \sum_i p_i^* \mathcal{L}_{reg}\left(t_i,\ t_i^*\right) \tag{1}$$

In Equation (1), *i* is the index of an anchor. When the total loss function is evaluated, it can be observed that the loss in the sibling layers for each anchor box is normalized by the number and added.

Herein, $\lambda$ is a parameter that balances the regression and classification terms. The imbalance of each term occurs because $N_{cls}$ indicates the minibatch size and $N_{reg}$ indicates the number of anchors.

$\mathcal{L}_{cls}$ is the log loss for two classes and is expressed according to Equation (2). Additionally, $p_i$ is the probability that an object is present at the $i$th anchor, and $p_i^*$ represents the ground-truth label (1 or 0).

$$\mathcal{L}_{cls}\left(p_i, p_i^*\right) = -\left(p_i^* \log(p_i) + \left(1 - p_i^*\right) \log(1 - p_i)\right) \tag{2}$$

In Equation (1), $\mathcal{L}_{reg}$ is the loss calculation for the regression of the box. If the label $p_i^*$ of the ground-truth box is zero, it interrupts the entire training. Thus, it is necessary to remove the loss for this case by adding a calculation that multiplies $p_i^*$. $\mathcal{L}_{reg}$ is defined as Equation (3), where R is expressed according to Equation (4),

$$\mathcal{L}_{reg}\left(t_i, t_i^*\right) = R\left(t_i - t_i^*\right) \tag{3}$$

$$\text{smooth L1 loss } R(x) = \begin{cases} 0.5x^2, & |x| < 1 \\ |x| - 0.5, & otherwise \end{cases} \tag{4}$$

where $t_i$ and $t_i^*$ in Equation (3) are four-dimensional vectors used to compare the center coordinates of the $i$th anchor box with the center coordinates of the predicted box and ground-truth box, respectively. That is, each vector consists of a comparison of the $x$ coordinate, $y$ coordinate, width, and height of the center of the box. Accordingly, each vector can be defined according to Equations (5) and (6) listed below.

$$t_x = (x - x_a)/w_a, t_y = (y - y_a)/h_a, t_w = \log(w/w_a), t_h = \log(h/h_a) \tag{5}$$

$$t_x^* = (x^* - x_a)/w_a, t_y^* = (y^* - y_a)/h_a, t_w^* = \log(w^*/w_a), t_h^* = \log(h^*/h_a) \tag{6}$$

where $x$ represents the $x$-coordinate of the center of the predicted box, $x_a$ is the $x$-coordinate of the center of the anchor box, and $x^*$ is the $x$-coordinate of the center of the ground-truth box. The same notation applies to $y$, $w$, and $h$. Based on the calculation of $\mathcal{L}_{reg}$, the predicted box can be regarded as a regression from the anchor box to the ground-truth box nearby.

### 3.5. Fast R-CNN Detector (RoI Pooler)

Region proposals generated by RPN and feature maps extracted from backbone CNN are used as the inputs of the Fast R-CNN detector structure. The Region of Interest (RoI) pooling layer used the RoI align layer suggested in Mask R-CNN [40]. That is, a fixed-length feature vector is extracted by aligning each object proposal with a feature map through the RoI align layer. As in RPN, the vector extracted in this way is used as an input to two fully connected layers (i.e., the regression and classification layers). Herein, unlike the classification layer of the RPN, the Fast R-CNN module determines what the object class is in the proposed region using a softmax function.

The loss derived through the above model is one of four types, that is, the box-regression layer loss and classification layer loss of the RPN and Fast R-CNN detector. The loss of the entire model is selected by aggregating these four loss types, and the loss function is optimized according to the selected optimizer. When observing the training by aggregating the loss of the RPN and Fast R-CNN detector, it can be observed that the training of the entire model is end-to-end. Herein, the optimizer used stochastic gradient descent (SGD) optimizer with momentum.

## 4. Experiments and Results

### 4.1. Dataset

This study aims to detect various defects in the exterior image data of buildings using a multi-object detection model. To this end, in this study, we first selected delamination, cracks, peeled paint, and

water leaks as typical defects that occur in the building façade through previous literature reviews and defined these defects as the detection target.

As can be observed in Table 1 below, there are 10,907 raw image datasets used in this study. After splitting the train and test datasets at a ratio of 8:2, we separated the validation dataset from the train dataset for the fine-tuning of the model. That is, the entire image dataset was separated into train, validation, and test datasets at a ratio of 7:1:2, based on the number of images. The entire image data were obtained using a digital camera and had a resolution of $4032 \times 1960$ pixels.

**Table 1.** Image dataset for Faster R-CNN.

|  | Category | Before Crop Augmentation | After Crop Augmentation | Validation Set | Test Set |
|---|---|---|---|---|---|
|  | images | 7635 | 157,584 | 1091 | 2181 |
|  | Class 1: delamination | 5201 | 104,073 | 659 | 1430 |
| objects | Class 2: crack | 5668 | 114,852 | 834 | 1717 |
|  | Class 3: peeled paint | 1234 | 25,267 | 165 | 348 |
|  | Class 4: leakage of water | 222 | 3773 | 38 | 47 |

Because of the large image size, the size of the feature map that is output from the convolutional layer of the model is also large. Accordingly, the number of parameters in the fully connected layer increases significantly. In the case where the number of parameters cannot be determined based on the total number of image samples, the model is likely to be overfitted to the train dataset. Hence, it is not generalized to other datasets apart from the training dataset, and results in poor performance. From the several existing regularizations, the input image size was resized to $800 \times 600$ to prevent such overfitting. In fact, when these images are resized to smaller sizes, the information is lost given that objects composed of lines, such as cracks, cannot be observed with the naked eye. The resized image indicates a state where information is lost from the original image data. However, we can reduce computational costs and prevent overfitting. Therefore, the entire image was fixed at a resolution of $800 \times 600$.

Because of the large architecture size of ResNet-50 used as a backbone in the Faster R-CNN structure, there are many parameters associated with the entire model. On the contrary, as the number of raw image data was small, data augmentation was performed on the training dataset. It was performed separately from the augmentation method in the model, and the random crop method was used for this. The random crop method usually refers to a random cropping operation using padding. However, in this study, we modified the random crop method as shown in Figure 6. First, a window of arbitrary size in the raw image was set. Subsequently, a random point was determined within the set window, and the raw image was cut out based on the horizontal and vertical lines that passed through this point. It was then resized to the previously set image size of $800 \times 600$. Herein, the object in the image (ground-truth box) can be cut off together with the image. If more than 50% of the ground-truth box remains before being cut off, the object is set to exist. Additionally, if the number of random points is set to k, new images generated through data augmentation are produced by multiplying the number of original image data by k times. In this study, the window size was set to $100 \times 100$ at the top left corner of the image, and k was set to 20. Further, we prevented overfitting with the use of horizontal flipping for data augmentation, as shown in Figure 7.

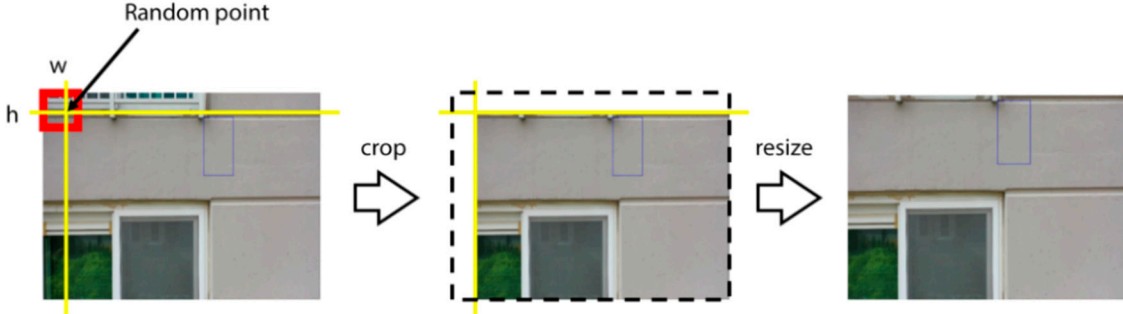

**Figure 6.** Process of data augmentation with random cropping (w = 100, h = 100).

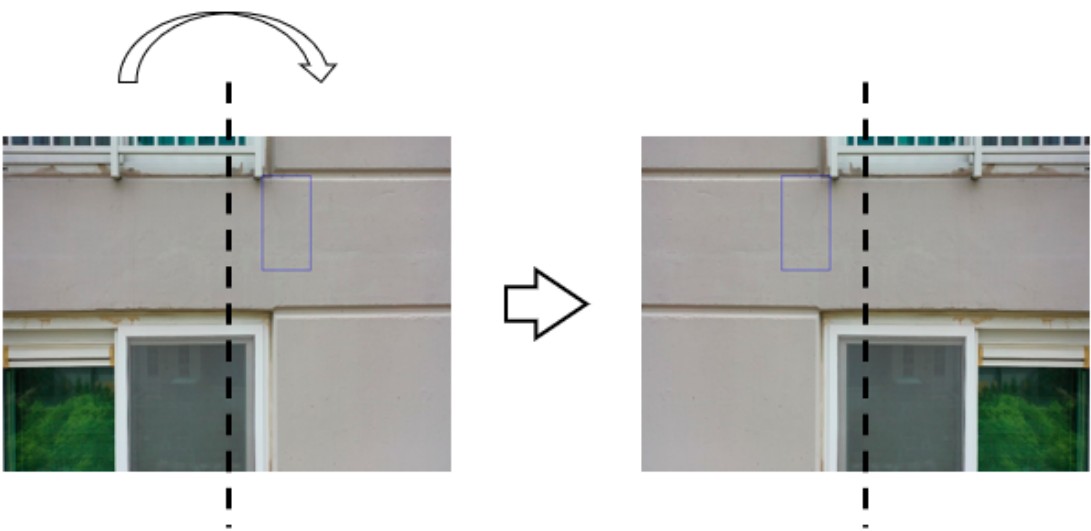

**Figure 7.** Process of data augmentation with horizontal flipping.

### 4.2. Results

In the Faster R-CNN network, there are many hyperparameters, such as the anchor scale, ratio, rescaling variable, and the non-maximum suppression (NMS) threshold. The main point in this study is the performance derived when Faster R-CNN is applied to multiclass defect detection tasks in the building façade. Therefore, the hyperparameters used in the traditional Faster R-CNN model were mainly used.

The optimizer used for model training is the SGD with momentum. This optimizer has a feature that reflects the direction of the previous weight update when the weights are updated. The weight update formula of SGD with momentum is expressed in the form of Equation (7), where $\beta$, $m$, and $\varepsilon$ refer to the momentum, momentum vector, and learning rate, respectively. In this study, the minibatch size for weight updates was set to eight, and $\beta = 0.9$ and $\varepsilon = 0.0001$ based on experiments.

$$w \leftarrow w - \beta m - \varepsilon \nabla_w L(w) \tag{7}$$

Figure 8 shows the loss curve derived when performing experiments based on the above settings. When the loss curve is checked, overfitting is considered to occur because the validation loss converges at approximately 13,000 iterations and gradually increases after this. Therefore, we derived the experimental results with the use of the early stopping technique.

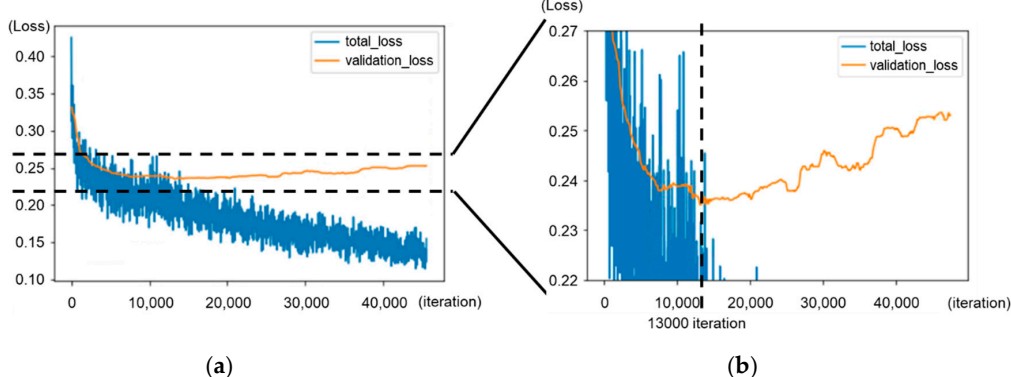

**Figure 8.** Learning curve. (**a**) Original scale image of learning curve; (**b**) y axis scaling image of learning curve.

The right side of Figure 8 confirms the increase of the validation loss after approximately 13,000 iterations. Based on this, we determined the loss at 13,000 iterations as the optimal value of the loss function and measured the performance of the model by loading the weight at that time.

The metric that determines the performance of the model is the average precision (AP) used in the COCO dataset. Each metric indicated the performance of object proposals and instance detection, and the precision was calculated using Equation (8).

$$Precision = \frac{TP}{TP + FP} \tag{8}$$

The AP calculation for the COCO dataset was calculated differently and according to IoU. AP50 is a calculated value when each IoU threshold is set to 0.5. The mean average precision (mAP) is the average of AP for IoUs that consisted of 0.5 to 0.95 at an interval of 0.05. The calculation results of the relevant indices for the test dataset are shown in Table 2, and a graph of AP results derived using the validation dataset for each iteration is shown in Figure 9. The output for the test set of the model trained in this study is shown in Figure 10.

**Table 2.** Results of AP50/mean average precision (mAP) for each class. IoU = intersection over union.

| Category | AP (IoU = 0.5) (%) | AP (IoU = 0.5:0.05:0.95) (%) |
|---|---|---|
| All Classes | 62.717 | 31.487 |
| Class 1 | 49.765 | 27.289 |
| Class 2 | 69.732 | 42.201 |
| Class 3 | 50.140 | 26.829 |
| Class 4 | 67.260 | 29.629 |

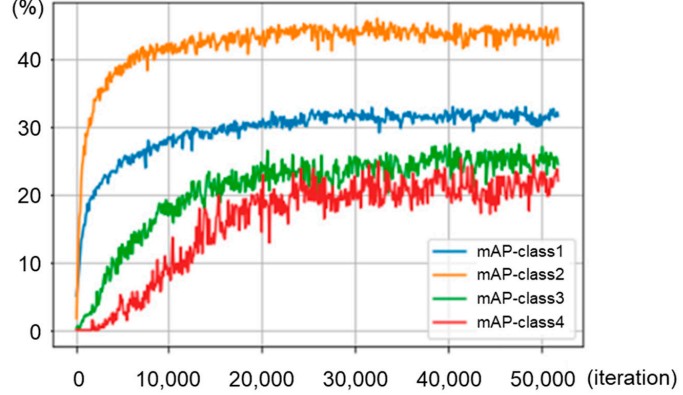

**Figure 9.** Results of mean average precision (mAP) for each class in validation set.

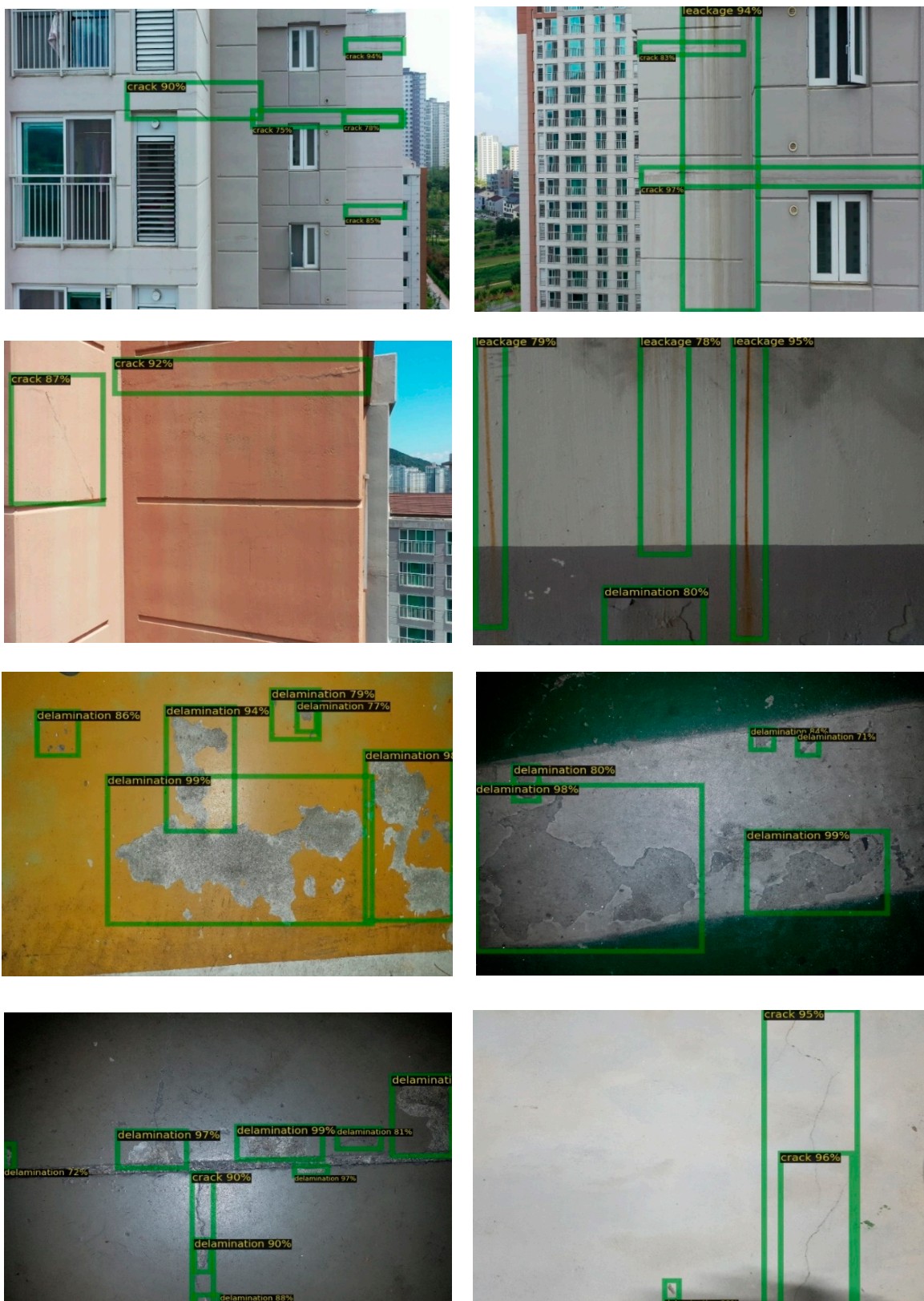

**Figure 10.** Examples of output model images.

As it can be observed in Table 2, the AP (IoU = 0.5) of the entire class was 62.7%, and the highest AP was confirmed to be cracks (Class 2). It was followed by leakage (Class 4) at 67.3%, peeled paint (Class 3) at approximately 50%, and delamination (Class 1) at approximately 49.8%.

As shown in Figure 9, the mAP for Classes 3 and 4 undergo severe oscillation. This is considered to occur because of a few instances for these classes. That is, as observed in Table 1, the number of instances of Classes 3 and 4 is confirmed to be less than that of Classes 1 and 2. Basically, the database has a major influence on the performance of DCNN models. In particular, the database size and balance are crucial. From this perspective, it is considered that the performance of this model can be further improved if the instance of each class is expanded and the balance is taken into account.

## 5. Conclusions

Multiple defects occur in various locations in actual buildings. To minimize the negative effects of these defects on the sustainability of buildings, there is a need for a technology that can efficiently monitor defects. In this study, we aimed to simultaneously detect various types of defects in the building façade in the real world using the Faster R-CNN model. The application performance of this model was verified through the collected data, and the average performance of each defect type was approximately 60% based on AP (IoU = 0.5). This is considered to be a meaningful result, justifying the possibility of multiple defect detection using a deep learning model.

In fact, various studies exist that detect defects based on deep learning, but most of them detected a single defect in laboratory conditions or focused on civil structures. However, this study used real-world building defect image data. That is, it is not a dataset built under refined conditions. We used real-world data where irregularities always exist, and various image interferences occur. Furthermore, unlike civil structures, the building façade is composed of various shapes and colors. This means that the background irregularities of image data are quite common. Thus, detecting defects is very challenging.

In general, defect management in a building is a way of dealing with defects that have already occurred. However, this method has limitations in minimizing performance reduction and maintenance cost increase due to defects. To solve these problems, it is necessary to check defects frequently during the maintenance phase, but the existing manpower-oriented defect inspection method is costly. Unmanned defect inspection techniques associated with the model proposed in this study can solve these problems. However, in order to manage the building efficiently, a variety of additional technologies need to be developed along with the deep learning-based MultiDefectNet proposed in this study. In other words, various technologies, such as unmanned aerial vehicle technology, defect location detection technology, and durability assessment technology, need to be linked to each other in order to develop unmanned defect inspection technology.

Also, it is necessary to expand the database (DB) for training and make it well-balanced to increase the accuracy of the model proposed in this study. In addition, it is considered that there is a need for more research on CNN architecture or data preprocessing that can distinguish image backgrounds from defects based on the considerations of the characteristics of buildings that feature various façade shapes.

**Author Contributions:** K.L. developed the concept and drafted the manuscript. G.H. and L.S. reviewed the manuscript. S.L. and H.Y.K. supervised the overall work. All authors have read and agreed to the published version of the manuscript.

**Funding:** This research was supported by a grant (19CTAP-C152020-01) from Technology Advancement Research Program (TARP) funded by Ministry of Land, Infrastructure and Transport of Korean government.

**Acknowledgments:** The authors would like to thank the Ministry of Land, Infrastructure and Transport of the Korean government for funding this research project.

**Conflicts of Interest:** The authors declare no conflict of interest.

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
