# Peer review of "MultiDefectNet: Multi-Class Defect Detection of Building Façade Based on Deep Convolutional Neural Network"

_sustainability, doi:10.3390/su12229785_

Round 1

Reviewer 1 Report

The article is well structured and deals with an important topic.

The issue of building maintenance is increasingly topical, but the increasing complexity of the construction process, changes to sustainability parameters and globalisation make upgrading operations increasingly difficult. The authors are asked how it is possible to “develop a model capable of detecting multiclass defects from real-world image data obtained from various building facades” when there is an ever-increasing number of variables, with very different features?

The façade depends on the structure, the materials, the technologies, the quality of the installation, its appropriateness in the location, for the air pollution, the acoustic standards, etc.

Thinking about all the variables of an element (facade) is complex, even more so if you think about the building.

The whole must then be seen with the expected duration of the element and therefore of the building. All this must lead to a rethink of how to manage maintenance.

Comments and Suggestions for Authors

What methods had been implemented in this research?  

Could the study be aimed at a precise facade technology placed in a building that is architecturally and structurally defined and above all contextualised?

Author Response

Thank you for your comments. Please check the answer to your comments through the attached file.

Reviewer 2 Report

The article proposes an artificial neural network for detecting damage to building facades.

The authors focus on the detailed specification of the approach used. Little space has been devoted to the very nature of the type of damage and the circumstances of its occurrence. The conclusions could be broader with a clear emphasis on the practical value of the proposed approach and the limitations.

Author Response

(The authors gave the same response as above.)
